# Genome-Wide Identification and Expression Analysis of the Xyloglucan Endotransglucosylase/Hydrolase Gene Family in Sweet Potato [*Ipomoea batatas* (L.) Lam]

**DOI:** 10.3390/ijms24010775

**Published:** 2023-01-01

**Authors:** Jing-Zhen Zhang, Pei-Wen He, Xi-Ming Xu, Zun-Fu Lü, Peng Cui, Melvin Sidikie George, Guo-Quan Lu

**Affiliations:** 1College of Advanced Agricultural Sciences, Zhejiang A&F University, Hangzhou 311300, China; 2Crop Science Department, Njala University, Njala Campus, Private Mail bag, Freetown 999127, Sierra Leone

**Keywords:** *Ipomoea batatas*, xyloglucan endotransglucosylases/hydrolases (*XTH*s) gene family, gene-wide identification

## Abstract

The xyloglucan endotransglucosylase/hydrolase (XET/XEH, also named XTH) family is a multigene family, the function of which plays a significant role in cell-wall rebuilding and stress tolerance in plants. However, the specific traits of the *XTH* gene family members and their expression pattern in different tissues and under stress have not been carried out in sweet potato. Thirty-six XTH genes were identified in *I. batatas,* all of which had conserved structures (Glyco_hydro_16). Based on Neighbor-Joining phylogenetic analysis the *IbXTHs* can be divided into three subfamilies—the I/II, IIIA, and IIIB subfamilies, which were unevenly distributed on 13 chromosomes, with the exception of Chr9 and Chr15. Multiple cis-acting regions related to growth and development, as well as stress responses, may be found in the *IbXTH* gene promoters. The segmental duplication occurrences greatly aided the evolution of *IbXTHs*. The results of a collinearity analysis showed that the *XTH* genes of sweet potato shared evolutionary history with three additional species, including *A. thaliana*, *G. max*, and *O. sativa*. Additionally, based on the transcriptome sequencing data, the results revealed that the *IbXTHs* have different expression patterns in leaves, stems, the root body (RB), the distal end (DE), the root stock (RS), the proximal end (PE), the initiative storage root (ISR), and the fibrous root (FR), and many of them are well expressed in the roots. Differentially expressed gene (DEG) analysis of FRs after hormone treatment of the roots indicated that IbXTH28 and IbXTH30 are up-regulated under salicylic acid (SA) treatment but down-regulated under methyl jasmonate (MeJA) treatment. Attentionally, there were only two genes showing down-regulation under the cold and drought treatment. Collectively, all of the findings suggested that genes from the XTH family are crucial for root specificity. This study could provide a theoretical basis for further research on the molecular function of sweet potato *XTH* genes.

## 1. Introduction

The cell wall is a crucial characteristic feature of plant cells. Plant cell walls serve as an exterior supporting structure that greatly affects the size and form of the cells during the growth and development of the plant. The cell wall mostly consists of cellulose, hemicellulose, lignin, and pectin. Cell-wall reconstruction is accompanied by breakage and regeneration of the xyloglucan cell wall [1]. Xyloglucan, which is a cellulose chain containing oligosaccharide side chains, is the most important hemicellulose component in the primary cell wall of dicotyledons [2]. XTHs, one of the key enzymes in cell-wall remodeling, is widely present in plant cells; it can catalyze the connection and breakage of xyloglucan molecules and modify the fiber–xyloglucan composite structure by controlling the elasticity and ductility of cells [3].

The *XTH* family belongs to the glycoside hydrolase family 16 (GH16), genes in which is contained a typical enzymatic reaction motif (DEIDFEFLG). There are two catalytic activities, including xyloglucan endohydrolysis (XEH) and xyloglucan endotransglucosylases (XET) generally carried out by the *XTH* gene familywhich participate in the relaxation, synthesis, and breakdown of cell walls. Glyco_hydro_16 and XET_C are two conserved domains found in *XTH* proteins. Three subfamilies (I, II, and III) can be distinguished within the *XTH* family base, and subfamily III is divided into IIIA and IIIB subfamilies. Of those reported so far, the *XTH*s in the I/II subfamily and IIIB subfamily have been discovered to have considerable xyloglucan endotransglucosylases (XET) activity, whereas the *XTH*s in IIIA subfamily have xyloglucan endohydrolysis (XEH) activity.

*XTH*s are widely distributed in mosses, lycophytes, ferns, angiosperms, and gymnosperms, and the first *XTH* gene was discovered in cowpea in 1992 [4]. Nowadays, a total of 33, 43, 25, 56, 29, and 61 *XTH* genes were identified in *Arabidopsis thaliana* [5], *Populus simonii* × *Populus nigra* [6], *Solanum lycopersicum* [7], *Nicotiana tabacum* [8] *Oryza sativa* [9], and *Glycine max* [10] *XTH*s are vital for the growth and development of plants and showed a diversity of tissue expression. For instance, in *A. thaliana*, at least 10 genes are strongly expressed in the root differentiation zone promoting cell-wall synthesis, cell elongation and expansion. Five genes (*AtXTH1*, *AtXTH21*, *AtXTH22*, *AtXTH30,* and *AtXTH33*) are expressed in green siliques. *AtXTH24* and *AtXTH32* are expressed in stems, and five genes are expressed in stems [5,11]. At present, it has been discovered that *XTH* contributes to the development and strengthening of cell walls. Overexpression of *PttXET16-34*, which originated from a hybrid aspen (*Populus tremula* × *tremuloides*), promotes vascular growth and development and enhances xyloglucan levels in primary-walled xylem but lowers the level of xyloglucann secondary-walled xylem [12]. In addition, it has been demonstrated that the maize *ZmXTH1* gene controls the composition and structure of the cell wall [13] Recent research has revealed that *Arabidopsis XET*-related genes are also influenced by inanimate factors such as a haptic, dark, cold shock, heat shock, wind, floods, salinity, and light, as well as by metabolites such as ethylene, canola lactone, gibberellin, and auxin [14]. After being exposed to exogenous ethylene, the *XET* gene of kiwifruit is triggered to express. The expressions of *Arabidopsis XTH21* [8] and *OsXTH8* in rice [15] were up-regulated by gibberellin treatment. The *LeXTH* gene of tomato was up-regulated under the treatment of exogenous auxin (2,4-D) [16]. The *Arabidopsis AtXTH23* gene was significantly up-regulated under abscisic acid treatment [5].

Overexpression of the *CaXTH3* gene in pepper, which can be induced by high salt, drought, and cold, causes severe folding of *Arabidopsis* leaves [17]. The *DkXTH6* gene of persimmon was suppressed at low temperatures, whereas *DkXTH7* transcription was highly active [18]. In addition, *DkXTH1* overexpression has been shown to impact the growth of roots and leaves and improve the tolerance of transgenic *Arabidopsis* to salt, drought stress, and ABA treatment [19]. 

Sweet potato [*Ipomoea batatas* (L.) Lam.]—which is the seventh most significant food crop in the world, with high-yielding plantingand widespread farming in developing countries—is a hexaploid with 90 chromosomes [20]. Moreover,;it is also highly valued as food, feed, and medicine because of the tuberous root, stems, and leaves containing a wealth of proteins, lipids, anthocyanins, carotenoids, and so on [21].

This study used the genomes of *I. batatas* to carry out genome-wide identification and analysis of the *IbXTH* family genes. Furthermore, chromosome locations, gene structures, gene systems analysis, cis-acting element, Neighbor-Joining phylogenetic tree analysis, tissue-expression pattern, and stress expression characteristics were subsequently analyzed. The findings of this project will open up new avenues for research into the biological mechanisms underlying the evolution of *IbXTHs,* hence supplying crucial data for a profound thesis of the functions and advancement of this gene family in higher plants.

## 2. Results

### 2.1. Phylogenetic Analysis and Classification of IbXTHs

There are 36 *XTH* gene members in the sweet potato genome; these *XTH* gene members were designated *IbXTH01*–*IbXTH36* according to their position in the chromosome (Figure 1). In this study, the BLAST and HMMER search methods were used and then followed by identification using the Pfam tool. A total of 36 XTH genes were identified in the genome of *I. batatas* and named from *IbXTH*1 to *IbXTH*36 by their chromosome position. The chromosomal location of *IbXTHs* showed that the 36 identified XTH genes were unequally mapped on the 13 chromosomes, except for chromosomes 9 and 15. For instance, chromosome 13 contained the largest number of *IbXTHs* (8), while chromosomes 1, 5, 6, 8, and 12 only contained one *IbXTH* gene. Gene cluster exists for chromosome 3, 7, 13, and 14 and was also identified in the chromosome distribution map, especially in the interval 2523570 to 25228579 of Chr13 containing two gene clusters with three and five *IbXTHs,* respectively, indicating there are tandem duplication events in the *IbXTH* gene family.

To clarify the taxonomic and evolutionary relationships of the identified sweet potato *IbXTH* gene family, an Neighbor-Joining phylogenetic tree was constructed by MEGA7.0 with 36 XTH protein sequences from sweet potato and 33 AtXTH protein sequences (Figure 2). According to the previous subfamily classification of *Arabidopsis* XTH proteins—in which AtXTH1-AtXTH26 belong to subgroup I/II, AtXTH31 and AtXTH32 belong to subgroup III-A, and the rest of the proteins belong to subgroup III-B—sweet potato XTH proteins were classified into three subfamilies, namely I/II (blue), IIIA (yellow), and IIIB (pink). Among these, the I/II subfamily was the largest group, with 51 XTH protein sequences including 22 XTH genes in sweet potato There were eight XTHs in IIIA, and the IIIB subfamily in which XEH activity has been only reported contained 10 proteins.

### 2.2. Identification and Sequence Analysis of XTH Genes in I. batatas

Among the 36 IbXTH*s* in sweet potato, the shortest (IbXTH*13*) was of 164 aa, while the longest (IbXTH*8*) was of 344 aa, and the average length of genes was 280.22 amino acids. The predicted MWs of the *Ib*XTH were as high as 38.73763 kDa (IbXTH8) and as low as 3.30876 kDa (IbXTH17). The ***pI*** (isoelectric point) values of 21 out of 36 XTH in *I. batatas* were higher than 7.0, indicating that most of the sweet potato XTHs were alkaline proteins. The ***pI*** value of *Ib*XTH04 was only 4.69, whereas the ***pI*** value of IbXTH21 as the biggest was of 9.23, and others distributed between 4.69 and 9.38. There was a 50/50 split between stable proteins (<40) and unstable proteins. GRAVY (Grand average of hydropathicity) showed that all IbXTH proteins were hydrophilic, with an average GRAVY of −0.3595. Subcellular localization analysis showed that, 36.1% IbXTH proteins were extracellular, while 63.8% were predicted to be located in the plasma membrane. The aliphatic index of the peptides ranged from 57.91 (IbXTH*18*) to 101.33 (IbXTH*13*) (Table 1).

### 2.3. Gene Structure Analysis of Sweet Potato XTHs

Generally, genes clustered in a subfamily have similar structures. In this research, XTHs in the I/II subfamily had one to three introns, except for IbXTH08 and IbXTH23 with four introns and IbXTH15 (zero introns). There are three to four introns in IIIA and IIIB subfamilies, except for IbXTH05, which has two introns belongs to the IIIA subfamily. 

To further explore the phylogenetic relationships of the IbXTH family and to investigate their potential functions, PFAM data were used to analyze the conserved domains of 36 XTH protein sequences. Results showed that the Glyco_hydro_16 domain was found in all IbXTHs, and 29 IbXTHs contained both the Glyco_hydro_16 and the XET_C domains, holding 80.56% in all genes. In addition, to further reveal the structural diversity of *IbXTH* protein sequences, MEME online software combined with the visualized motif pattern function of TBtools was used to analyze the related motifs. As shown in Figure 3 and Figure 4, the motif number of the IbXTH family members ranged from four to nine. The XET_ C domain mainly covered motif 5. Motif 4 and motif 5 were the longest motifs containing 41 aa, while the shortest contained only 10aa (motif 9). Motif 1 was widely present in I/II and IIIA and IIIB, which contains the active site DEIDFEFLG (E is the catalytic point). The IbXTH5, IbXTH12, and IbXTH34 had the highest number of motifs, with nine motifs which had a consistent arrangement order, starting with motif 10 and ending with motif 5. However, IbXTH13 and IbXTH31 had the least number of motifs, with only four motifs. Furthermore, we also found that motif 10 only existed in the IIIA subfamily, suggesting that motif 5 might have evolutionary specificity. Except for IbXTH9, all IbXTH gene members of the IIIA and IIIB subfamilies had motif 9.

### 2.4. Chromosomal Distribution and Synteny Analysis of IbXTHs

It was discovered that four pairs of *IbXTHs,* which belong to inter-chromosome segmental duplications, had a collinear (red) relationship between and within chromosomes in sweet potato based on the collinearity analysis performed by MCScanX (Figure 5). We inferred that segmental gene duplication might play a significant role in the growth of the *IbXTH* gene. Furthermore, the duplicated genes belonged to the same subfamily, and the four pairs of genes were found to have strong collinearity. *IbXTH18* and *IbXTH19*, *IbXTH17,* and *IbXTH32* were in the I/II subfamily; *IbXTH12* and *IbXTH34* were in the IIIB subfamily; and *IbXTH20* and *IbXTH21* belonged to the IIIB subfamily. In addition, in order to comprehend the mechanism of gene divergence and evolutionary pressure, the *Ka/Ks* ratios were also determined. The gene is more susceptible to nonsynonymous mutation if the *Ka/Ks* ratio is >1, which denotes positive selection. The ratio indicatesneutral selection if it is equal to 1, while indicated purifying selection. I Ifthe ratio is <1, and synonymous mutation is more likely in the gene. All Ka/Ks values were less than 0.5, meaning that XTH family genes have gone through a strong purifying selection during evolution (Table 2).

To further explore the gene duplication events of the XTH gene family, collinearity and replication analysis of all genes were carried out using TBtools software, and three comparative synthetic maps of three representative species with sweet potato were constructed, including two dicotyledonous plants (*A. thaliana*, *G. max*) and one monocotyledonous plant (*O. sativa*) (Figure 6). The collinearity analysis results showed that a total of 63 covariate pairs were found, including 12 pairs in *A. thaliana* (green line); 49 XTH gene pairs exits in *G. max* (purple line); and only one pair in *O. sativa* (red line) compared to *I.batatas*. In addition, *IbXTH24* showed a high collinear relationship with three comparative species (two in *A. thaliana*, four in *G. max*, and one in *O. sativa*), suggesting that *IbXTHs* showed higher evolutionary differentiation in dicotyledonous plants than in monocotyledonous plants.

### 2.5. Cis-Acting Elements in IbXTHs

To predict the function of *IbXTH* gene family, the 2000bp upstream CDS sequences were extracted from the promoter regions of 36 IbXTH*s* of sweet potato by using PlantCARE database (http://bioinformatics.psb.ugent.be/webtools/plantcare/html/ (accessed on 15 September 2022). It was found that the promoter region of the sweet potato XTH family is abundant in cis-acting elements, which are divided into the following six categories: light-response, hormonal-response, environmental-stress-related, developmental-related, binding-site-related, and other elements (Figure 7). All XTHs of sweet potato contained 5 to 19 light-responsive elements, including the GTGGC-motif, TCCC-motif, and TCT motif. Five hormonal-related response elements were invested in all IbXTH*s*, ABA(ABRE), GA (GARE-motif, TATC-box, and P-box), IAA (AuxRR-core, TGA-element, and TGA-box), MeJA (CGTCA-motif and TGACG-motif), and SA (TCA-element and SARE). Endosperm expression (GCN4_motif), meristem expression elements (CAT-box), and seed-specific regulation (RY-element) are development-related elements, which were also determined to exist in half of the IbXTH gene. Among the elements related to environmental stress (Figure 7c), anaerobic induction (ARE) and low-temperature responsiveness (LTR) was found in 31 IbXTH*s*. In addition, there are 42 MYB response elements bonded with drought-inducing ability (MBS), flavonoid biosynthetic (MBSI), and light responsiveness (MRE).

### 2.6. Expression Profiles of XTH Genes in Sweet Potato

To investigate the potential biological functions of *IbXTHs* in the growth and development of sweet potato, the expression profiles of the 36 *IbXTHs* were detected in seven different tissues by transcriptome data of “Xushu 18” retrieved from the GEO database (PRJNA511028). The expression of *IbXTHs* in each tissue of sweet potato was obtained, and a heat map was drawn according to FPKM (Fragments Per Kilobase of exon model per Million mapped fragments). If the FPKM of a gene is less than 1, it is considered that the gene lacks expression data.In this study, The FPKM of*IbXTH*5, *IbXTH*7, *IbXTH*9, *IbXTH*13, *IbXTH*15, and *IbXTH*36 in all tissues were less than 1, indicating that these genes had no function or didn’t expressed. On this basis, the remaining 30 *IbXTHs* were expressed in at least one tissue (Appendix A).

The *IbXTHs* had different expression patterns in various tissues (Figure 8a), which implied that the *IbXTH*s might have played different roles in developmental processes. Generally, almost half of the identified *IbXTHs* (14 genes) in sweet potato were unregulated in all of the detected tissues. The others were highly expressed in the root body (RB), distal end (DE), root stock (RS), proximal end (PE), and initiative storage root (ISR), while they were downregulated in the leaf, stem, and fibrous root (FR), which showed strong tissue-specific expression. Compared to other family members in the XTH gene family of sweet potato, it is noteworthy that *IbXTH*3, *IbXTH*22, *IbXTH*28, *IbXTH*29, and *IbXTH*30 have relatively high expression levels. Among them, *IbXTH*28, *IbXTH*29, and *IbXTH*30 have high expression levels in root tissues other than FR, indicating that these genes may be necessary for the development of root tissues.

Furthermore, the FB expression patterns of *IbXTHs* were analyzed under various abiotic stresses and different hormone treatments (Figure 8b). Nine *IbXTHs* were identified not expressed in all the treatments tested, including *IbXTH*5, *IbXTH*7- *IbXTH*9, *IbXTH*11, and *IbXTH*13–*IbXTH*15. On this basis, the remaining 27 *IbXTHs* were expressed in at least one treatment. As shown in Figure 8b (Appendix A), there were 10 differential expression genes (DEGs) in the FR after SA treatment (9 up-regulated and 1 down-regulated), 13 after MEJA treatment (7 up-regulated and 6 down-regulated), and only 1 down-regulated gene in FR under the treatment of ABA(*IbXTH0*2). It is worth mentioning that the *IbXTH*28, *IbXTH*29, and *IbXTH*30 were the most significantly up-regulated under MEJA treatment. FKPM expression had increased by about 72, 104, and 112 times, respectively. In addition, we analyzed the differential expression genes (DEGs) of *IbXTHs* under various abiotic stresses, such as salt, drought, and cold stress. The *IbXTH* gene family members of the FR of sweet potato had no up-regulated genes under all abiotic stresses but had two down-regulated genes, including *IbXTH0*2 and *IbXTH*12, which were down-regulated in cold and drought stress. *IbXTH*12 was also down-regulated under the stress of salt. 

### 2.7. Tissue-Specific and Stress-Inducible Expression of IbXTH02 and IbXTH12

Subsequently, the expression patterns of two *IbXTHs* (*IbXTH02* and *IbXTH12*), which displayed substantial change in the RNA-data and which down-regulated under salt and drought treatment and showed xyloglucan endo-hydrolase activity, were further examined under two abiotic stresses: salt stress (0.2 mol/L NaCl) and PEG-induced drought stress (20% PEG6000) by qRT-PCR assay, and a two-fold cut-off value was explored (Appendix A, Figure 9). Under PEG-induced drought stress (Figure 9a,b; Appendix A), *IbXTH*02 and *IbXTH*12 in roots were down-regulated in all periods, while the expression of *IbXTH*02 and *IbXTH*02 were expressed at a high level at 0 h and showed a trend of first decreasing and then increasing after treatment. The expression level of *IbXTH*12 reached the lowest level at 12 h, which was down-regulated with 604-fold induction. The expression level of *IbXTH*12 reached the lowest level at 24 h, which was down-regulated with 153-fold induction. *IbXTH0*2 and *IbXTH*12 were also down-regulated under salt stress (Figure 9c,d; Appendix A). It is worth mentioning that *IbXTH*02 had the lowest relative expression at 12 h after salt treatment (0.0042), with 236-fold induction down-regulated. Moreover, *IbXTH*12 displayed a tendency of first declining, then increasing, and then decreasing. At 24 h of salt treatment, it had decreased by nearly 267-fold, reaching its lowest expression level. These results indicated that different genes had different response times to drought stress in sweet potato. Collectively, these results implied that *IbXTH* genes might have different expression under abiotic stress. 

## 3. Discussion

Xyloglucan in the cell wall participates in the reconstruction, breakage, and regeneration of the cell wall. The XTH gene family has been identified in several plant species, including *Arabidopsis* [5], rice [9], barley [22], poplar [23], and tomato [4,7].

In this study, we report the identification and characterization of the sweet potato *XTH* gene family and constructed a phylogenetic tree using the XTH proteins from sweet potato and *Arabidopsis*. Expression pattern analysis suggested that *IbXTHs* may play an important role under various stresses, in which *IbXTH02* and *IbXTH12* showed down-regulation under PEG6000 and NaCl treatment.

### 3.1. Charaterization of Sweet Potato XTHs Gene Family

It is well-known that plant XTHs play a vital role in regulating the extensibility of the cell wall, but there is limited information on the number of XTH gene family members in sweet potato and the evolutionary relationships between *IbXTH* genes. The number of *IbXTH* genes discovered was significantly lower than in *Nicotiana tabacum* [24] and *Glycine max* [10] and was grouped into three categories by previous phylogenetic studies in *Arabidopsis.* Furthermore, evolutionary mechanism analysis suggested that the *IbXTHs* family expanded partly due to segmental duplication events, which further lead to the conserved protein motif and gene structure. These *IbXTH*s genes showed different expression patterns in different tissues and six treatments. For example, *IbXTH17* exhibited the highest expression level in the initiative storage root (ISR), while *IbXTH32* expressed highest in the stem. *IbXTH20* and *IbXTH21* showed the highest expression in the root stock (RS), but lowest in the fibrous root (FR). The expression patterns of these collinear gene pairs implied that the sweet potato *XTH* gene family might have shown new function during the subsequent evolution. With that considered, the discovery of the *IbXTH* gene family lays the foundations and offers important insights for further study of the sweet potato *XTH* gene family.

### 3.2. IbXTH02 and IbXTH12 Gene Expression in Abiotic Stress

More and more evidences point to the role of *XTHs* as major players in modifying how plants react to various harmful environmental stimuli. In response to drought stress, previous studies have shown that a plant would close the stomates first to prevent water evaporation, and *XTH* genes are essential for modifying the cell wall’s elongation and improving drought resistance [25]. For instance, virus-induced gene silencing of *HvXTH1* in barley demonstrated that *BSMV: HvXTH1* plants showed an increase in shoot fresh weight and a reduction in water loss compared with BSMV:γ plants under drought stress [26]. The *AtXTH18* gene was silenced by the RNAi method, and the elongation of the first rooted epidermal cells was inhibited. *XTH3* from *Capsicum annuum* enhanced drought resistance of transgenic *Arabidopsis thaliana* and *S. lycopersicum* when the seedlings of which were exposed to extreme drought stress [27]. Moreover, it is notable that the existence of xylglucan in early terrestrial plants implies that members of the XTH gene family played a crucial role in the transition from wetter to drier habitats [28]. To better understand the possible function of *IbXTH* genes under PEG-induced drought stress, we also analyzed in detail the expression of *XTH* genes in sweet potato. The results showed that the *IbXTH02* and *IbXTH12* expressed were down-regulated in FR.

Under salt stress, a previous studies reported that *MtXTH3* in *Medicago truncatula* was strongly up-regulated at higher NaCl concentration [29]. In our research, the NaCl-induced salt treatment and RT-qPCR analysis showed that *IbXTH02* and *IbXTH12* were significantly down-regulated, suggesting that the IbXTH02 and IbXTH12 might have similar functions. Taken together, these findings provide novel information about *IbXTHs* under abiotic stress, especially drought and salt stresses. It can be speculated that the above genes may show increased cell wall-related functions under stress by combining with xyloglucan. Nevertheless, further molecular and genetic identification efforts are needed to verify their functions.

## 4. Materials and Methods

### 4.1. Genome-Wide Identification of XTH Genes

The *I. batatas* genome sequencing database was derived from *I. batatas* ‘Taizhong 6’ genomic data, which were provided by the Ipomoea Genome Hub (https://sweetpotao.com [30] (accessed on 10 September 2022)). The genome sequences and general feature format files of *Arabidopsis thaliana* were downloaded from TAIR (http://www.arabidopsis.org/ [31] (accessed on 10 September 2022)). The reference sequence was taken as the XTH protein sequence of *Arabidopsis thaliana*, which was downloaded from the *Arabidopsis* Information Resource (https://www.arabidopsis.org/ [31] (accessed on 10 September 2022)). This sequence was used as the query sequence to scan by using the Protein Basic Logical Alignment Search Tool (BLASTP, United States National Library of Medicine) with an E-value (≤1 × 10^−5^) and an identity match (≥50%) as thresholds of TBtools (https://github.com/CJ-Chen/TBtools [32] (accessed on 18 September 2022)). The Hidden Markov Model (HMM) profiles of the XET_C (PF06955) domain and Glyco_hydro_16 (PF00722) domain were derived from the Pfam database (https://pfam.xfam.org/ [33] (accessed on 18 September 2022)). Thesedomains were used as baits to conduct a Hidden Markov Model (HMM) with an e-value < e^−10^ searching for sequence homologs by using HMMER 3.3.1 [33] (http://hmmer.org/download.html [34] (accessed on 18 September 2022)). The candidate XTH genes were identified from the integration of BLASTP and Hmmer search results and then submitted to the NCBI’s conserved domains database (https://www.ncbi.nlm.nih.gov/cdd/ [35] (accessed on 19 September 2022)).

### 4.2. Physicochemical Property Analysis of IbXTH Protein Sequence

The XTH proteins of basic physicochemical properties—including a number of amino acids, Molecular Weight (MW), protein length, isoelectric point (PI), grand average of hydropathy (GRAVY), and instability index—were predicted by using the online tool of ProtProm in ExPASy (https://web.expasy.org/protparam/ [36] (accessed on 20 September 2022)). At last, the subcellular localization of XTH proteins was calculated by using Euk-mPLoc 2.0 (http://www.csbio.sjtu.edu.cn/bioinf/euk-multi-2/ [37] (accessed on 21 September 2022)).

### 4.3. Chromosomal Localization of XTH Genes

The positions of *IbXTHs* on chromosomes were obtained from the sweet potato genome annotation information, which was obtained from the Ipomoea genome hub (https://sweetpotao.com/ [31] (accessed on 10 September 2022), and chromosome mapping was performed and visualized by using TBtools [32].

### 4.4. Conserved Motif and Gene Structure Analyses

The MEME online tools (http://meme-suite.org/ [33] (accessed on 18 September 2022)) were used to identify the conserved motif, with ten discovery motif numbers and other default parameters. The gff3 file of the *I. batatas* genome contains information about the gene structure, which can be used to identify the exons and introns of XTH. The conserved motif and gene structure can be visualized by TBtools [32].

### 4.5. Phylogenetic Tree Construction

The XTH protein sequences of *A. thaliana* (At) and *I. batatas* were aligned by using Clustal Musle [38], with the default parameters used to generate a neighbor-joining (NJ) phylogenetic tree [39] using MEGA 7.0 software [40]. The tree was constructed with 1000 replicates bootstrap analysis, the Jones–Taylor–Thornton (JTT) model, and the missing data treatment option set at partial deletion and beautified by Chiplot (https://www.chiplot.online/ (accessed on 30 September 2022)) online tools. The XTH gene subfamily of *I. batatas* was categorized based on the *A. thaliana* XTH gene subfamily.

### 4.6. Collinearity Analysis and Ka/Ks Calculation

The duplication types and intraspecific covariance of XTH family members were analyzed using the One Step MCSanX-Super Fast function in TBtools [32]. The Advanced Circle function in TBtools [32] was adopted to plot the collinearity relationship of XTH genes, and the collinearity relationship between *I. batatas* and three other species, including *G. max*, *O. sativa,* and *A thaliana*, and the genome sequences and general feature formats of which were derived from Ensemble Plants (https://plants.ensembl.org/index.html [41] accessed on 20 September 2022).

### 4.7. Cis-Acting Elements Analyses

The 2000 bp promoter sequences of the sweet potato XTH gene were submitted to the PlantCare website (http://bioinformatics.psb.ugent.be/webtools/plantcare/html/ [42] (accessed on 25 September 2022)) for cis-acting regulation element prediction based on the *I. batatas* genome database file. The results of the PlantCare analysis were simplified and then visualized by using TBtools [32].

### 4.8. Expression Analysis of XTH Genes

To explore the expression patterns of sweet potato XTH genes, the RNA-seq expression profiles of the XTH genes were mined from the sequence read archive (SRA) of NCBI under the project of PRJNA511028, which was a transcriptome analysis of the different tissues and stress and hormone treatments to the fibrous root (FR) of *I. batatas* ‘Xushu18’. The heat maps of the expression levels of the XTH genes were visualized using the Heatmap illustrator program in the toolkit of TBtools [32].

### 4.9. Plant Materials and Abiotic Stresses

The cultivated sweet potato variety ‘Xushu18’ was used in this study. The seedlings of this variety were collected from Zhejiang A&F University, China. The two cultivars’ uniform seedlings were raised and naturalized in the Hoagland solution at 26 °C with a 16/8 photoperiod for 3 days. The seedlings were treated with 0.2 mol/L NaCl and 1/2 Hoagland solution containing 20% PEG6000 mass/volume fraction, respectively, to simulate drought and salt abiotic stress. The fibrous root collected after 0 (CK), 3, 6, 12, 24, and 48 h.

Total RNA from all the collected fibrous root was extracted using SteadyPure Plant RNA Extraction Kit (, Accurate Biotechnology, Hunan, China.) in accordance with the manufacturer’s instructions to validate the RNA-seq data. Reverse transcription was performed on 1 μg of RNA from each sample using the Evo M-MLV RT Mix Kit with gDNA Clean for qPCR (Accurate Biotechnology, Hunan, China). The qRT-PCR assay was conducted by a CFX Connect Real-Time System (Bio-Rad, Veenendaal, UT, USA) using the SYBR Green Premix Pro Taq HS qPCR Kit Accurate Biotechnology, Hunan, China). The specific primer of gene *IbXTH02* and *IbXTH12* for qRT-PCR detection are designed by Primer6 software listed in the Appendix A. The sweet potato *β-Actin* gene (GenBank, AY905538) was applied as the internal control. The experiments were conducted for three replicates for each gene, and the 2^−∆∆CT^ method was used to calculate the results [33]. We analyzed the data and compared the means using LS at a 0.01 level of significance.

## 5. Conclusions

We identified 36 IbXTH genes and then analyzed their physicochemical properties. Based on the phylogenetic analysis, the IbXTHs were divided into three subfamilies. We also investigated their promoter regions and collinearity relationships with three species. In addition, we systematically investigated the expression profiles in different tissues, and different development stages of storage roots, as well as the expression of the *IbXTHs* under NaCl-indued salt stress and PEG-induced drought treatment. The *IbXTH*-mediated stress-response mechanism in sweet potato can be furtherresearched due to the diversification of the *IbXTH* genes.

## Figures and Tables

**Figure 1 ijms-24-00775-f001:**
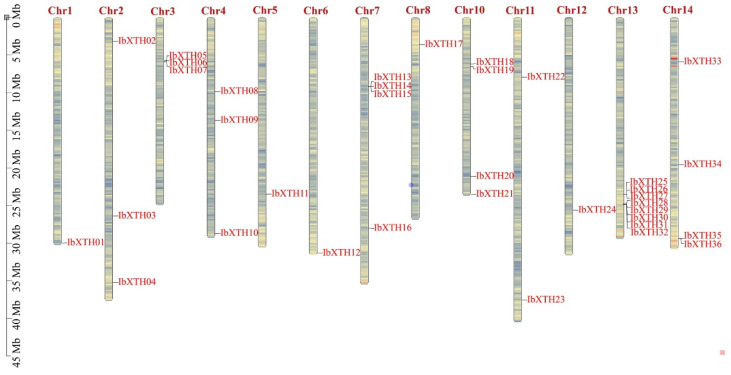
Gene location of the sweet potato *XTHs*, which were designated based on their chromosomal locations. The map was constructed and visualized by TBtools.

**Figure 2 ijms-24-00775-f002:**
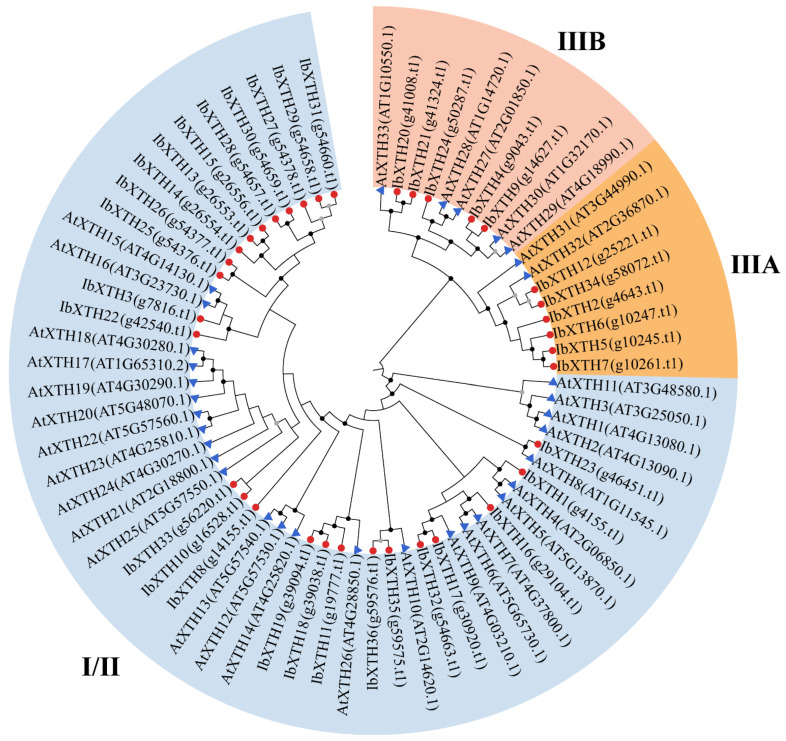
Neighbor-Joining phylogenetic tree analysis of XTH proteins from sweet potato and *Arabidopsis* constructed by MEGA 7.0 with the default parameters. The evolutionary tree was divided into three subfamilies with different colors. According to the previous subfamily classification of *Arabidopsis* XTH proteins, in which AtXTH1 -AtXTH26 belong to subgroup I/II, AtXTH31 and AtXTH32 belong to subgroup III-A, and the rest of the proteins belong to subgroup III-B, sweet potato XTH proteins were classified into three subfamilies, namely I/II (blue), IIIA (yellow), and IIIB (pink). The blue triangle represents *Arabidopsis* XTH proteins, while the red circle represents sweet potato XTH proteins.

**Figure 3 ijms-24-00775-f003:**
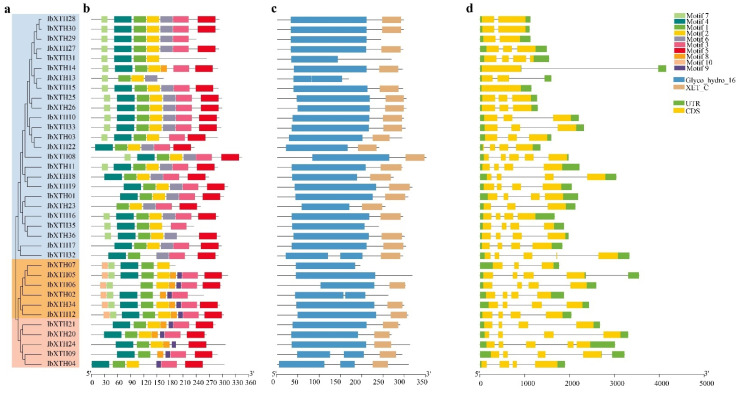
Characterization of the sweet potato XTH genes. (**a**) Phylogenetic tree of sweet potato XTH genes family. Genes from groups IIIA and IIIB are shown in the orange and pink color, respectively. Subfamily I/II is shown blue. (**b**) Protein conserved motifs constructed by the MEME Boxes with different colors represented different conserved motifs. (**c**) Conserved protein domain.The blue box represents Glyco_hydro_16 domain, while the brown boxes represent XET_C domain (**d**) Gene structures. Yellow boxes represent exons; green boxes indicate the UTRs; and black lines represent introns.

**Figure 4 ijms-24-00775-f004:**
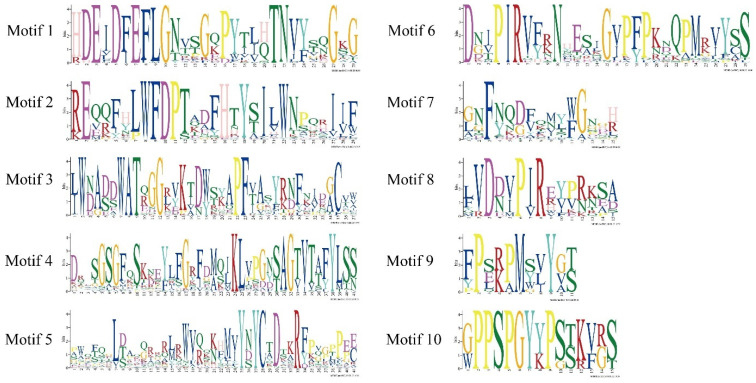
Schematic diagram of the core motifs of the sweet potato XTHs (Appendix A); the diagram was created by MEME online program. The X-axis shows the conserved sequences of the motifs, and the conservation of residues are indicated by the height of the letters. The Y-axis represents the conservation of the amino acid.

**Figure 5 ijms-24-00775-f005:**
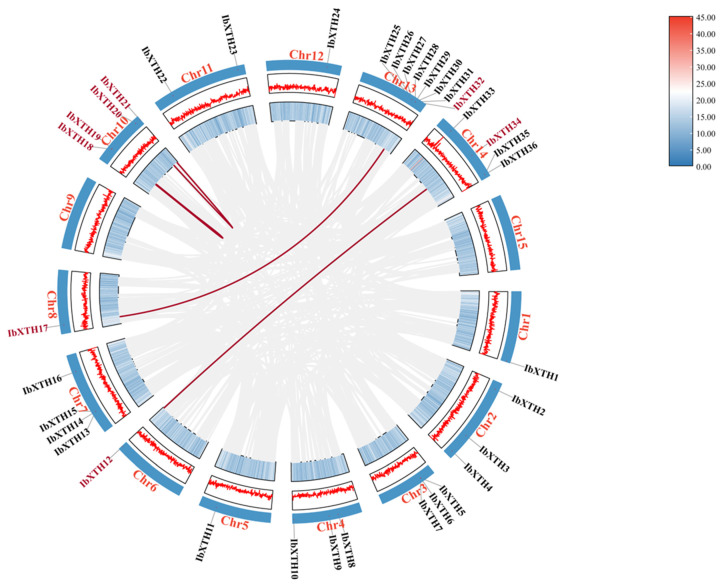
Fragment duplication analysis of the sweet potato XTHs. The red lines represent the fragment duplicate gene pair, and the gray lines represent the synteny blocks of the XTH genes in the sweet potato genome. The line and heat map in the outer circle represent gene density on the chromosome.

**Figure 6 ijms-24-00775-f006:**
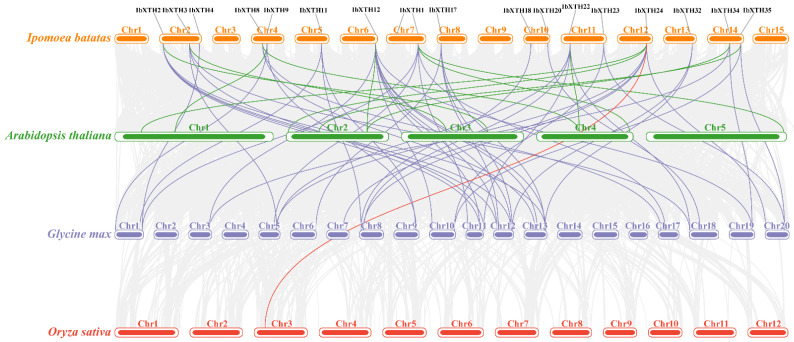
Collinearity analysis of the XTHS from sweet potato and three other species which were analyzed and visualized by Tbtools. The green lines represent XTH syntenic gene pairs between sweet potato and *Arabidopsis*. The purple lines represent XTH syntenic gene pairs between sweet potato and bean. The red lines represent XTH syntenic gene pairs between sweet potato and rice, and the gray lines represent orthologous genes of sweet potato with three other species.

**Figure 7 ijms-24-00775-f007:**
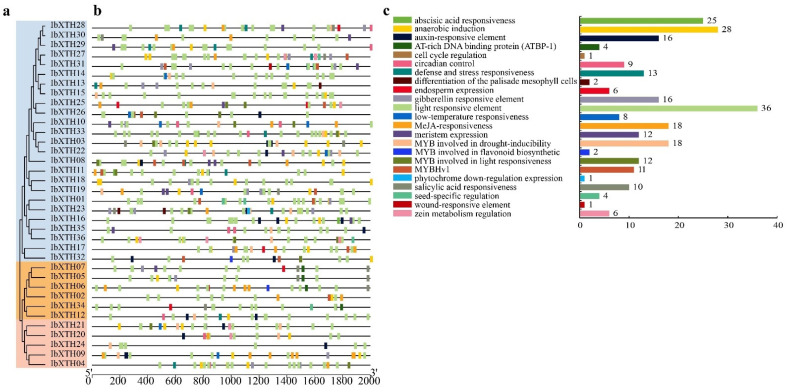
Cis-elements predication in the 2 kb sequences upstream of the sweet potato XTH gene promoters. (**a**) The phylogenetic tree of the sweet potato XTH genes clustered by MEGA7.0 Genes from groups IIIA and IIIB are shown in the orange and pink color, respectively. SubfamilyI/II is shown in blue. (**b**) Cis-acting element distribution, each box filled with a different color represents different promoters. (**c**) Cis-acting element gene numbers.

**Figure 8 ijms-24-00775-f008:**
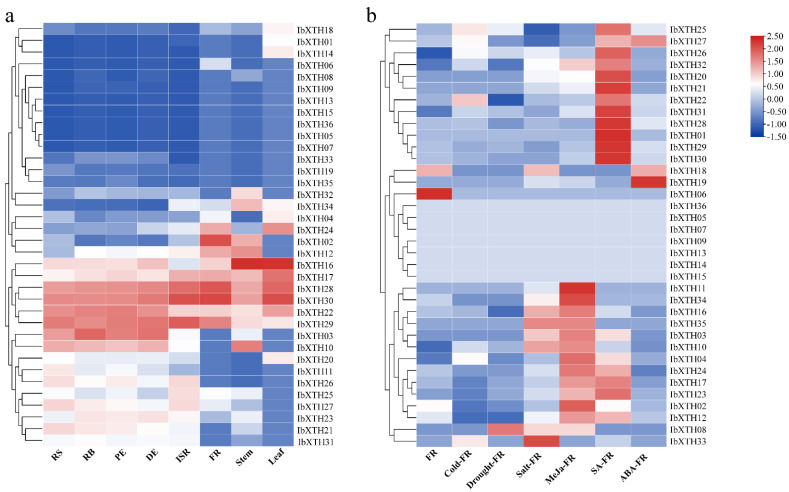
Expression pattern of XTH genes in different tissues and different treatments. (**a**) Expression pattern of the sweet potato XTH genes in different tissue with col scale. RS represents root stock, RB represents root body, PE represents proximal end, DE represents distal end, ISR represents initiated storage root, and FR represents fibrous root. (**b**) Expression pattern of the sweet potato XTHs in FR tissue under different abiotic stress and hormone treatment with row scale. Red represents high expression, and blue represents low expression. All ratios are log_2_ transformed so that inductions and repressions of identical magnitude are numerically equal but opposite in sign.

**Figure 9 ijms-24-00775-f009:**
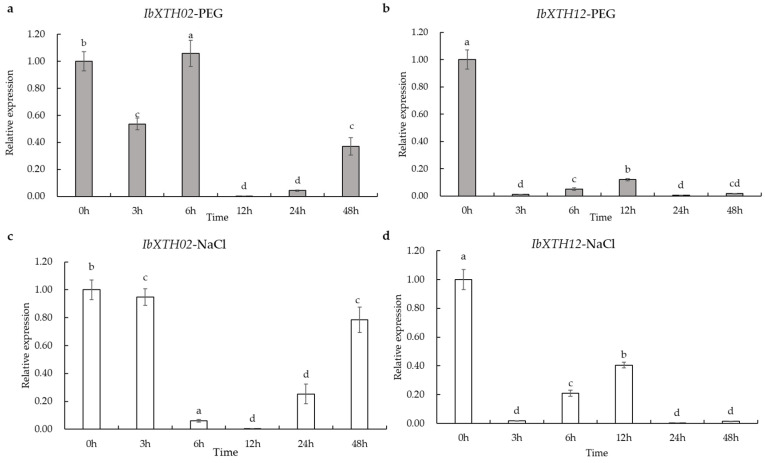
Expression profiles of *IbXTH02* and *IbXTH12* in response to PEG6000 and NaCl treatment (**a**) Expression patterns of *IbXTH02* in response to PEG-induced drought stress in FR of Xushu 18 were examined by a qPCR assay. (**b**) Expression patterns of *IbXTH12* in response to PEG-induced drought stress in FR of Xushu18 were examined by a qPCR assay. (**c**) Expression patterns of *IbXTH02* in response to NaCl stress in FR of Xushu18 were examined by a qPCR assay. (**d**) Expression patterns of *IbXTH12* in response to NaCl stress in FR of ‘Xushu 18’ were examined by a qPCR assay. The fibrous root was obtained at 0, 3, 6, 12, 24, and 48 h after PEG6000 and NaCl stress. The mean fold changes of each gene between treated and control fibrous root in Xushu18 were used to calculate its relative expression levels. Data are means ± SD of three biological replicates. Means denoted by the same letter do not significantly differ at *p* < 0.05, as determined by Duncan’s multiple range test.

**Table 1 ijms-24-00775-t001:** Basic information and physicochemical properties of sweet potato XTH genes.

Gene Name	ID	Chr	Chr. Location	Sub-Family	Gene Length (bp)	No. of Extron	MW (Da)	ProteinLength (aa)	*pI*	GRAVY	Position of Signal Peptide	Subcellular Localization
IbXTH01	g4155.t1	1	29990981–29993169	I/II	2188	6	32711.61	312	8.78	−0.233	1–26	Plasma membrane
IbXTH02	g4643.t1	2	3189988–3191861	IIIA	1873	6	11385.8	101	8.37	−0.579		Extracellular
IbXTH03	g7816.t1	2	26403969–26405559	I/II	1590	5	26222.77	234	7.7	−0.183	1–25	Plasma membrane
IbXTH04	g9043.t1	2	35246607–35248505	IIIB	1898	7	33429.48	294	4.69	−0.245	1–26	Extracellular
IbXTH05	g10245.t1	3	5774294–5777845	IIIA	3551	5	35208.95	302	8.84	−0.463	1–29	Plasma membrane
IbXTH06	g10247.t1	3	5793450–5796045	IIIA	2595	6	28209.47	255	8.72	−0.472	1–20	Extracellular
IbXTH07	g10261.t1	3	5916674–5918440	IIIA	1766	7	35378.8	303	9.07	−0.703		Plasma membrane
IbXTH08	g14155.t1	4	9862595–9864572	I/II	1977	7	34343.94	312	9.03	−0.34	1–23	Plasma membrane
IbXTH09	g14627.t1	4	13704162–13707389	IIIB	3227	6	33361.75	296	6.19	−0.338	1–19	Extracellular
IbXTH10	g16528.t1	4	28722021–28724227	I/II	2206	5	23192.44	211	9.2	−0.255	1–19	Plasma membrane
IbXTH11	g19777.t1	5	23479883–23482102	I/II	2219	6	20828.56	191	7.09	−0.205	1–23	Plasma membrane
IbXTH12	g25221.t1	6	31346152–31348193	IIIA	2041	6	38737.63	344	8.62	−0.515		Extracellular
IbXTH13	g26553.t1	7	9166497–9168089	I/II	1592	3	33099.43	288	6.33	−0.335	1–22	Extracellular
IbXTH14	g26554.t1	7	9172874–9177026	I/II	4152	4	32722.77	292	6.51	−0.337	1–30	Plasma membrane
IbXTH15	g26556.t1	7	9179941–9181089	I/II	1148	3	33184.97	289	5.27	−0.379		Plasma membrane
IbXTH16	g29104.t1	7	28028874–28030541	I/II	1667	6	34386.89	302	9.17	−0.346	1–26	Extracellular
IbXTH17	g30920.t1	8	3618290–3620130	I/II	1840	6	18725.45	164	5.5	−0.146	1–21	Extracellular
IbXTH18	g39038.t1	10	6180717–6183761	I/II	3044	6	32907.36	289	6.23	−0.283	1–22	Plasma membrane
*IbXTH*19	g39094.t1	10	6555568–6557616	I/II	2048	6	32602.98	290	6.65	−0.258	1–21	Plasma membrane
IbXTH20	g41008.t1	10	21143172–21146481	IIIB	3309	6	33303.48	290	6.31	−0.461	1–26	Extracellular
IbXTH21	g41324.t1	10	23509092–23511770	IIIB	2678	6	30330.31	268	9.23	−0.447		Extracellular
IbXTH22	g42540.t1	11	7946462–7947812	I/II	1350	5	29784.55	263	7.67	−0.398		Plasma membrane
IbXTH23	g46451.t1	11	37577929–37580059	I/II	2130	7	31835.92	283	7.08	−0.352		Plasma membrane
IbXTH24	g50287.t1	12	25631860–25634874	IIIB	3014	7	27305.22	235	8.89	−0.48		Extracellular
IbXTH25	g54376.t1	13	23523570–23524844	I/II	1274	5	29162.54	249	4.8	−0.545	1–28	Plasma membrane
IbXTH26	g54377.t1	13	23528294–23529584	I/II	1290	5	34922.22	306	6.51	−0.468	1–20	Plasma membrane
IbXTH27	g54378.t1	13	23534544–23536034	I/II	1490	4	33378.37	298	8.8	−0.422	1–29	Plasma membrane
IbXTH28	g54657.t1	13	25189323–25190449	I/II	1126	5	33342.38	298	8.8	−0.394	1–29	Plasma membrane
IbXTH29	g54658.t1	13	25192834–25193961	I/II	1127	4	32655.69	291	8.66	−0.408	1–23	Plasma membrane
IbXTH30	g54659.t1	13	25198284–25199394	I/II	1110	6	32967.05	292	8.48	−0.397	1–23	Plasma membrane
IbXTH31	g54660.t1	13	25202414–25203956	I/II	1542	4	26892.41	239	7.76	−0.184	1–23	Extracellular
IbXTH32	g54663.t1	13	25225244–25228579	I/II	3335	6	32882.88	292	8.54	−0.382	1–23	Plasma membrane
IbXTH33	g56220.t1	14	5902284–5904604	I/II	2320	6	28799.85	263	5.82	−0.039	1–23	Plasma membrane
IbXTH34	g58072.t1	14	19576311–19578744	IIIA	2433	6	32590.46	290	6.33	−0.448		Extracellular
IbXTH35	g59575.t1	14	29382354–29384228	I/II	1874	6	33154.24	296	6.89	−0.379	1–29	Plasma membrane
IbXTH36	g59576.t1	14	29385211–29387188	I/II	1977	5	33666.08	293	8.53	−0.397	1–22	Plasma membrane

**Table 2 ijms-24-00775-t002:** The Ka/Ks analysis of XTH homologous gene pair in sweet potato.

seq1	seq2	Ka	Ks	Ka/Ks
*IbXTH*18	*IbXTH*19	0.07	0.1865	0.3755
*IbXTH*21	*IbXTH*20	0.0174	0.075	0.2319
*IbXTH*32	*IbXTH*17	0.1747	1.2778	0.1367
*IbXTH*34	*IbXTH*12	0.1714	0.8654	0.1981

## Data Availability

The genome sequences of the sweet potato, including the predicted gene model annotation for this study, can be found in the Ipomoea Genome Hub (http://sweetpotato.com/). The RNA-Seq datasets referring to different tissues and different treatments of the sweet potato can be acquired from SRA of NCBI (PRJNA511028).

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
