# Peer review of "Genome-Wide Identification and Expression Analysis of the Xyloglucan Endotransglucosylase/Hydrolase Gene Family in Sweet Potato [Ipomoea batatas (L.) Lam]"

_ijms, 2023, doi:10.3390/ijms24010775_

Round 1
Reviewer 1 Report
In my opinion it is an interesting manuscript. The authors have attempted to analyse an important problem. The paper results are and quite interesting. “Introduction” section is well written and, in the final part, it presents points that authors want to highlight in the paper.
Having familiarized myself with the manuscript, I have some suggestions:
1. Add section Material and methods.
2. How the groups were distinguished (e.g. Fig. 2, Fig. 7).
3. Were the results statistically analyzed? If so, please describe the method.
Author Response
Response to Reviewer 1 Comments
Dear Editor and Reviewer:
On behalf of my co-authors, we are very grateful to you for giving us an opportunity to revise our manuscript. We appreciate you very much for your positive and constructive comments and suggestions on our manuscript entitled ‘Genome-Wide Identification and Expression Analysis of the Xyloglucan endotransglucosylase/hydrolase Gene Family in Sweetpotato [Ipomoea batatas (L.) Lam]. (ID:IJMS-2038469) ’. We have studied reviewers’ comments carefully and tried our best to revise our manuscript according to the comments. The following are the responses and revisions I have made in response to the reviewers' questions and suggestions on an item-by-item basis. Thanks again to the hard work of the editor and reviewer!
Point1. Add section Material and methods.
Thanks for your comment. We add section Material and methods.in line 476.
Point 2. How the groups were distinguished (e.g. Fig. 2, Fig. 7).
Thanks for your comment.The groups were distinguished by the result of FIG1. According to the previous subfamily classification of Arabidopsis XTH proteins, where AtXTH1- AtXTH26 belong to subgroup I/II, AtXTH31 and AtXTH32 belong to subgroup III-A and the rest of the genes belong to subgroup III-B, sweetpotato XTH proteins were classified into three subfamilies, named â… /â…¡(blue), IIIA (yellow), â…¢B(pink).
- Were the results statistically analyzed? If so, please describe the method.
Thanks for your suggestion. We followed this suggestion to revise.We make qPCR assay which was added for statistical analysis(line495-497).\
In general, we think your opinion is constructive. Thanks again for your suggestions.

Reviewer 2 Report
The manuscript is interesting and have good information. But it does not comprised any real experimental data. The authors analyzed the gene family very well but this is in silico work. The expression analysis also done on previous data generated. It required the validation of in silico expression through qRT PCR analysis.
Author Response
Response to Reviewer 2 Comments
Dear Editor and Reviewer:
Thank you for your decision and constructive comments on my manuscript entitled ‘Genome-Wide Identification and Expression Analysis of the Xyloglucan endotransglucosylase/hydrolase Gene Family in Sweetpotato [Ipomoea batatas (L.) Lam].(ID:IJMS-2038469) ’. We have studied reviewers’ comments carefully and tried our best to revise our manuscript according to the comments. The following are the responses and revisions I have made in response to the reviewers' questions and suggestions on an item-by-item basis. Thanks again to the hard work of the editor and reviewer!
Point 1.
It required the validation of in silico expression through qRT PCR analysis.
We think this is an excellent suggestion. The seedlings of sweetpotato ‘Xushu18’ were treated with salt and drought treatment. The specific primer of gene IbXTH2 and IbXTH12 for qRT-PCR detection and the sweetpotato β-Actin gene(GenBank,AY905538) was applied as the internal control. The experiments were conducted for three biological replicates for each gene and the 2−△△CT method was used to calculate the results We analyzed the data and compared the means using LS at a 0.01 level of significance.
In general, we think your opinion is constructive. Thanks again for your suggestions.

Reviewer 3 Report
The research paper titled “Genome-Wide Identification and Expression Analysis of the Xyloglucan endotransglucosylase/hydrolase Gene Family in Sweet potato [Ipomoea batatas (L.) Lam]” reports an important Xyloglucan endotransglucosylase/hydrolase (XTH) gene family which is having an important role in cell wall reconstruction and stress responses in plants. In this paper, authors performed a genome-wide LbXTHs investigation on sweet potato (Ipomoea batatas) and elucidated their function to abiotic stress in silico analysis.
However, the paper needs serious review and answer following comments.
1. Line 73 and 82. Those two paragraphs harp away at the same issues. The response of the XTH gene family to biotic, abiotic, and hormone treatments can be written in separate paragraphs.
2. Line 105. The paragraph titled ‘’ Phylogenetic Analysis of XTH Proteins’’ is hard to understand. Divide it into more sentences.
3. Line 119. Legend of Figure 1 needs to have more description for subfamilies. For example, the names and their colors have to be described.
4. Line 151 (and check the entire manuscript). Please re-write the gene name in italic form.
5. Line 159. Please put more descriptions in the legend of figure 2A, B, C and D. For example, write how to generate the evolutionary tree.
6. Line 207 and 231. Put the information about how this data was analyzed and visualized in the legend of figures 5 and 6.
7. The most important question to authors that why there is no qRT-PCR analysis after all those in silico analyses.
8. Please describe and conclude the most important gene(s) for further molecular or breeding research to create/develop new varieties tolerant to various abiotic and biotic stresses.
Author Response
Response to Reviewer 3 Comments
Dear Editor and Reviewer:
Thank you for your decision and constructive comments on my manuscript entitled ‘Genome-Wide Identification and Expression Analysis of the Xyloglucan endotransglucosylase/hydrolase Gene Family in Sweetpotato [Ipomoea batatas (L.) Lam].(ID:IJMS-2038469) ’. We have studied reviewers’ comments carefully and tried our best to revise our manuscript according to the comments. The following are the responses and revisions I have made in response to the reviewers' questions and suggestions on an item-by-item basis. Thanks again to the hard work of the editor and reviewer!
Point 1:
Line 73 and 82. Those two paragraphs harp away at the same issues. The response of the XTH gene family to biotic, abiotic, and hormone treatments can be written in separate paragraphs.
Response 1:
Thank you for the suggestion of my repetition. These duplicates are removed and consolidated into one piece of content.
Point 2:
Line 105. The paragraph titled ‘’ Phylogenetic Analysis of XTH Proteins’’ is hard to understand. Divide it into more sentences.
Response 2:
Thank you for your valuable comments for the paragraph title. The precedent version of the title has been replaced, becoming “Phylogenetic Analysis and classification of IbXTHs” (line107)
Point 3:
Line 119. Legend of Figure 1 needs to have more description for subfamilies. For example, the names and their colors have to be described.
Response 3:
We sincerely thank the reviewer for careful reading, and we modified the problem, (Line123-125) ----Figure 1. NJ phylogenetic tree analysis of XTH proteins from sweetpotato and Arabidopsis con-structed by MEGA 7.0 with the default parameters. The evolutionary tree was divided into 3 subfamilies with different colour. Genes from groups â…¢A and IIIB are shown in the orange and pink color, respectively. Group â… /â…¡was showed blue. The blue triangle represents Arabidopsis XTH genes while the red round represents sweetpotato XTH genes.
Point 4:
Line 151 (and check the entire manuscript). Please re-write the gene name in italic form.
Response 4:
We were sorry for our careless mistakes, thankyou for your reminder. As suggested by the reviewer, we have corrected the gene name in italic form to make the word harmonized within the whole manuscript.
Point 5:
Line 159. Please put more descriptions in the legend of figure 2A, B, C and D. For example, write how to generate the evolutionary tree.
Response 5:
Thanks for your valuable comments. We added the description of the figure(line167-174).
Figure 2. Characterization of the sweetpotato XTH genes. (a) Phylogenetic tree of sweetpotato XTH genes family, Genes from groups â…¢A and IIIB are shown in the orange and pink color, respectively. Subfamilyâ… /â…¡was showed blue. (b) Protein conserved motifs constructed by the MEME Boxes with different colors represented different conserved motifs. (c) Conserved protein domain, (d) Gene structures. The blue box represented Glyco_hydro_16 domain while the brown boxes represented XET_C domain (d) Gene structures. Yellow boxes represented exons, green boxes indicated the UTRs, while black lines represented introns.
Point 6:
Line 207 and 231. Put the information about how this data was analyzed and visualized in the legend of figures 5 and 6.
Response 6:
Thanks for your valuable comments. We added the description about the analyze and the visualization of the figure (line230, line255).
Line230: Figure 5. Collinearity analysis of the XTHS from sweetpotato and other three species which were analyzed and visualized by Tbtools. The green lines represent XTH syntenic gene pairs be-tween sweetpotato and Arabidopsis. The purple lines represent XTH syntenic gene pairs between sweetpotato and bean. The red lines represent XTH syntenic gene pairs between sweetpotato and rice, and the gray lines represent orthologous genes of sweetpotato with other three species.
Figure 6. Cis-elements predication analysis in the 2kb sequences upstream of the sweetpotato XTH gene promoters. (a) the phylogenetic tree of the sweetpotato XTH genes clustered by MEGA7.0, (Genes from groups â…¢A and IIIB are shown in the orange and pink color, respective-ly. Subfamilyâ… /â…¡was showed blue. (b) Cis-acting element distribution, each box filled with dif-ferent color represent different promoters. (c) Cis-acting element gene numbers.
Point 7:
The most important question to authors that why there is no qRT-PCR analysis after all those in silico analyses.
Response 7:
We think this is an excellent suggestion. The seedlings of sweetpotato ‘Xushu18’ were treated with salt and drought treatment. The specific primer of gene IbXTH2 and IbXTH12 for qRT-PCR detection and the sweetpotato β-Actin gene(GenBank,AY905538) was applied as the internal control. The experiments were conducted for three biological replicates for each gene and the 2−△△CT method was used to calculate the results We analyzed the data and compared the means using LS at a 0.01 level of significance.
Point 8:
Please describe and conclude the most important gene(s) for further molecular or breeding research to create/develop new varieties tolerant to various abiotic and biotic stresses.
Thanks for your valuable comments. We followed this suggestion to revise.(Line 518):Taken together, these IbXTH genes especially IbXTH02 and IbXTH12, perform crucial roles in the signal transduction pathways that are connected to abiotic stressors. which provides a certain theoretical support for the comprehensive and systematic study of the mechanism of IbXTH gene response to abiotic stress.
We tried our best to improve the manuscript and made some changes marked in red in revised paper which will not influence the content and framework of the paper. We appreciate for Editors/Reviewers' warm work earnestly, and hope the correction will meet with approval. Once again, thank you very much for your comments and suggestions.

Round 2
Reviewer 2 Report
The manuscript is now ok but still some minor revision required as bellow.
Line-88: “DkXTH6” should be italic.
Line-95: “Sweet potato” should be write as “sweet potato”.
Line-109: Do not start with digit.
Line 113-114: “Arabidopsis” should be in italic.
Line-431: “Chou and Shen, 2010” should be in numerical order.
Line 456: Correct the name of O. sativa.
Line 469. Put the space before [48].
Line 487: Add the number of supl. Table.
Author Response
Dear Editor and Reviewer:
Thank you for your decision and constructive comments on my manuscript entitled ‘Genome-Wide Identification and Expression Analysis of the Xyloglucan endotransglucosylase/hydrolase Gene Family in Sweetpotato [Ipomoea batatas (L.) Lam]. (ID:IJMS-2038469) ’. We have studied reviewers’ comments carefully and tried our best to revise our manuscript according to the comments. The following are the responses and revisions I have made in response to the reviewers' questions and suggestions on an item-by-item basis. Thanks again to the hard work of the editor and reviewer!
Point 1.
Line-88: “DkXTH6” should be italic.
We were sorry for our careless mistakes, thank you for your reminder. As suggested by the reviewer, we have corrected the gene name in italic form
Point 2.
Line-95: “Sweet potato” should be written as “sweet potato”.
Oops, this is our negligence. Thanks for your comment. We have corrected it.
Point 3.
Line-109: Do not start with digit.
Thanks for your suggestion. We have corrected it. This part is revised as:
There are 36 XTH gene members sweetpotato genome and which were named IbXTHIbXTH01–IbXTHIbXTH36 according to their position in the chromosome.
Point 4.
Line 113-114: “Arabidopsis” should be in italic.
Oops, this is our negligence. Thanks for your comment. We have corrected it in the whole manuscript.
Point 5.
Line-431: “Chou and Shen, 2010” should be in numerical order.
Oops, this is our negligence. Thanks for your comment. We have corrected it.
Point 6.
Line 456: Correct the name of O. sativa.
Oops, this is our negligence. Thanks for your comment. We have corrected it.
Point 7.
Line 469. Put the space before [48].
Thanks for your comment. We have corrected it in the whole manuscription.
Point 8.
Line 487: Add the number of supl. Table.
Thanks for your comment. We have added supl. Table.S6 in line 487. This part is revised as:
The specific primer of gene IbXTH02 and IbXTH12 for qRT-PCR detection are designed by Primer6 software listed in the Table S6.
In general, we think your opinion is constructive. Thanks again for your suggestions.

Reviewer 3 Report
Thank you for your response.
Author Response
Thank you for your decision and constructive comments on my manuscript entitled ‘Genome-Wide Identification and Expression Analysis of the Xyloglucan endotransglucosylase/hydrolase Gene Family in Sweetpotato [Ipomoea batatas (L.) Lam]. (ID:IJMS-2038469) ’.